# Near-Miss Symmetric Polyhedral Cages

**Bernard M. A. G. Piette *** and **Árpad Lukács**

Department of Mathematical Sciences, Durham University, Durham DH1 3LE, UK
* Correspondence: b.m.a.g.piette@durham.ac.uk

**Abstract:** Following the experimental discovery of several nearly symmetric protein cages, we define the concept of homogeneous symmetric congruent equivalent near-miss polyhedral cages made out of P-gons. We use group theory to parameterize the possible configurations and we minimize the irregularity of the P-gons numerically to construct all such polyhedral cages for $P = 6$ to $P = 20$ with deformation of up to 10%.

**Keywords:** uniform polyhedra; polyhedral cages; Platonic group; near-miss cages; Cayley graph; protein cage; nano-cage

## 1. Introduction

Recently, an artificial protein structure, referred to as TRAP-cage, was engineered from TRAP [1–4]. (TRAP is an acronym for "trp RNA-binding" attenuation protein. It is an 11 subunit RNA-binding protein that regulates expression of genes involved in tryptophan metabolism (trp) in Bacillus subtilis).

The structure was made out of 24 nearly regular hendecagons (11 edge polygon) each having 5 neighbors with which to share an edge. The remaining 6 edges per face define the boundary of 38 holes; 32 of them are triangles, 3 faces contributing 1 edge each, while the remaining 6 are in between 4 hendecagons, each contributing 2 of their edges to them. More recently, similar nearly regular structures made out of the same protein were identified [5,6].

The geometrical structure modeling the cage proteins described above is called a polyhedral cage, or p-cage for short [7]. A p-cage corresponding to the TRAP-cage with regular polyhedra is mathematically impossible; it can only be realized approximately, if the edge lengths and angles of the polygonal faces are slightly deformed. Such an object will be called a near-miss p-cage.

Engineering polyhedral nano-structures is not new and not restricted to proteins. For example, experimental techniques called DNA origami have been created by bio-engineers [8–10]. Such DNA structures differ mostly from protein cages in that they are mostly hollow as the DNA strands span the edges of the polyhedra that are created. Nevertheless, the regular or nearly-regular geometries we identify in our paper could be useful for a range of nano-structures including DNA origami.

The concept of chemical cage is also not new and they are observed or made in a number of contexts [11–15]. Polyhedral structures in chemistry are also common [11,16–18].

In the present paper, we apply graph theory to describe the connectivity of p-cages, at a level where whole molecules forming the faces of the cage can be modeled with planar polygons. We note that graph theory also plays an important role at the atomic level, in molecules [19–27] and also in nano-structures [28–35].

The aim of this paper is to identify these new geometries as new mathematical objects, but we also aim to help bio-nano engineers to identify the type of protein assemblies likely to form nano cages, such as the ones identified in [1–4].

A first study of p-cages was recently performed [7] by creating p-cages where all the faces are equivalent from a connectivity point of view. This resulted in a large number

of p-cages with a deformation below 10% and most of them, though not all, appeared to have faces with identical deformations. This suggests that one should consider p-cages where the faces are all equivalent modulo a congruent automorphism. The aim of this paper is to build such p-cages. In [7] it was shown that the connectivity between the faces of all *equivalent* p-cages can be constructed from the planar graphs of regular solids. All such p-cages were then constructed using polygonal faces ranging from hexagons up to polygons with 17 edges.

In this paper we build homogeneous symmetric congruent equivalent near-miss p-cages with deformation up to 10%, but unlike in [7] we impose a symmetry on the p-cages to ensure congruence between the faces. Aside from obtaining p-cages which are, in some sense, more symmetric, imposing the symmetry leads to a more efficient method to build the p-cages as there are fewer variables to adjust to optimise the p-cage near-symmetry. As the number of variables is much smaller than in the case of [7] we are able to consider all p-cages made out of faces ranging from hexagons up to polygons with 20 edges.

The paper is organised as follows. We start with a few formal definitions and then recall how the planar graphs of the regular solids are linked to the connectivity between the faces of equivalent p-cages. For each regular solid, we identify all the transitive subgroups of their symmetry group which we then use to determine which hole shapes are compatible with the equivalence of the faces.

For each regular solid, we use their symmetry group to derive a parametrization of the face vertices compatible with the equivalence and we use computer programs to identify the least deformed p-cages for each possible configuration. We finish by describing the obtained p-cages and presenting the images of some of the least deformed ones.

## 2. Symmetric Polyhedral Cages

Following [7] we define a polyhedral cage as an assembly of planar polygons, referred to as faces, separated by holes which are not required to be either planar or regular (see Figure 1). The edges of the polygonal faces are adjacent to either another face or a hole and will be referred to in what follows as shared-edges and hole-edges, respectively. In other words, every edge must either belong to two faces or belong to a single face as well as a hole. For any two adjacent edges at least one of them must be adjacent to a hole. Each face must also have at least three neighbors implying that the p-cage faces must have a minimum of six edges.

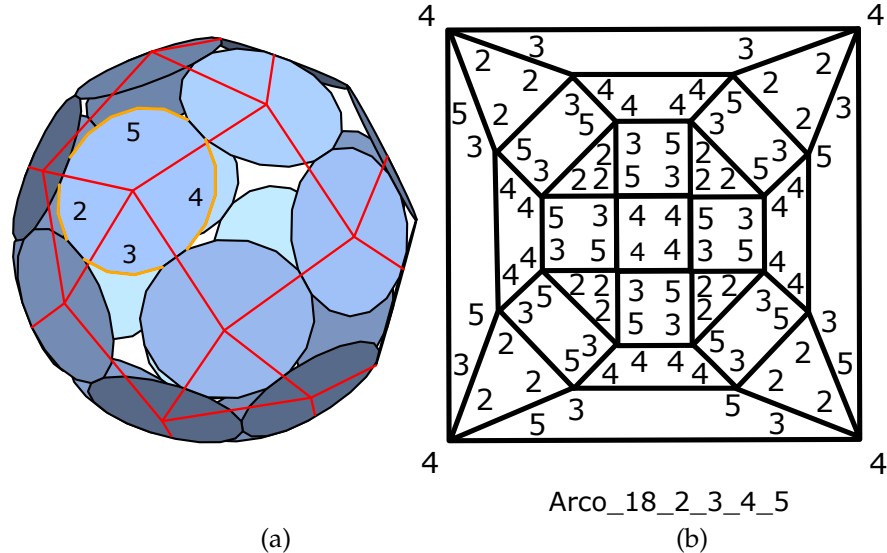

Arco_18_2_3_4_5

(a)                                                                 (b)

**Figure 1.** (**a**): schematic construction of the hole polyhedra of a p-cage. The number of hole-edges of the top left face is given for each hole to which the face contributes (the hole-edges for that face are orange on the figure). (**b**): the corresponding planar graph with the number of hole-edges around each vertex.

A homogeneous p-cage is one where all the faces are polygons with the same number of edges. As we shall only consider homogeneous p-cages in this paper, this qualifier will be omitted from now on.

A symmetric p-cage is defined as one for which for each pair of faces there is a congruent automorphism (a proper rotation) of the p-cage that maps one face onto the other. This implies that all the faces are identical.

The hull of a p-cage is defined as the polyhedron formed by the intersection of all the planes containing the faces [7]. A p-cage is said to be convex if its hull is convex. In what follows we will only consider convex p-cages.

The p-cages studied in [7] differ in that the faces are related by non congruent automorphisms, the equivalence between faces being only at the graph level or, in other words, the symmetry is at the connectivity level only and the faces can be different.

If the faces of the symmetric p-cages are regular polygons, the p-cage is said to be regular, but if the faces are not regular, the p-cage is described as near-miss. The deformation of near-miss p-cages can be so small that it is not noticeable to the naked eye, but it can also be very large. We shall define a measure of the amount of deformation below and restrict ourselves to deformations below 10%.

In what follows, we shall use $N$ to denote the number of faces of the p-cage and $P$ to denote the number of edges of the faces. Each hole will be made out of $Q_h$ edges where we include a hole index $h$ as a p-cage can have different types of holes.

### 3. Symmetric P-Cage Construction

As described in [7], if one joins the center of the face of the p-cage that shares one edge, one obtains a polyhedron, the face of which is not necessarily planar, but the resulting graph is a planar graph [36] (see Figure 1). We call it the hole-polyhedron because, by construction, its faces correspond to the holes of the p-cage. The vertices correspond to the faces of the p-cage and the edges to the links between the p-cage faces. This is effectively the dual of the p-cage and it encapsulates its connectivity.

Equivalence between the face of the p-cage is translated onto the hole-polyhedron as an equivalence between the vertices. Graphs for which all the vertices are equivalent are called Cayley graphs [37] and Maschke proved [38] that the only planar Cayley graphs are the graphs of the regular solids, i.e., the prism, antiprism, Platonic solids, and Archimedean solids. To construct a cage, one must choose $P$, the number of edges of a polygon as well as one of the Cayley graphs. A polygon is then placed on each vertex of the graph and linked to the adjacent faces according to the edges of the graph. One then has some flexibility to distribute the hole-edges in different ways between the corners around each vertex. For example, if one places an octagon on a trivalent vertex, such as on a tetrahedron (Figure 2), there are three shared edges and five hole-edges which must be distributed between the three adjacent faces of the Cayley graph. This can be done as 1,1,3 or 1,2,2, plus permutations, but this must also be done for each face of the p-cage so that the faces of the p-cage are all equivalent.

Formally, given a Cayley graph where each vertex has $d$ neighbors and $P$-gonal faces, the numbers, $q_i$ ($i = 1 \ldots d$), of hole-edges on each corner around a vertex must satisfy $\sum_{i=1}^{d} q_i = P - d$. Equivalence between the faces of the p-cage, which translates into equivalence between the vertices of the hole-polyhedron graph, implies that the sequence $q_i$ must be identical for all vertices up to a cyclic rotation which is also determined by the equivalence. As we shall see, for some p-cages, the invariance imposes that some pairs of $q_i$ must be identical. Also, like in [7], we do not consider p-cages with $q_i > P/2$, as such p-cages would not look like closed structures.

Before we proceed with the construction, it is useful to describe how we label the p-cages.

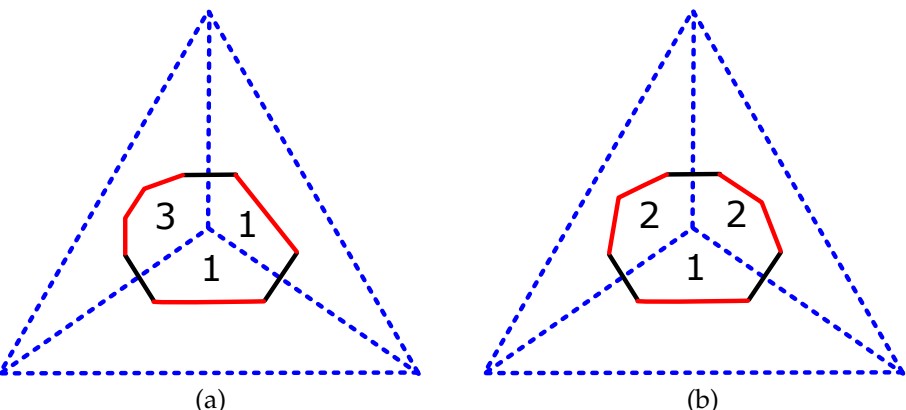

**Figure 2.** (**a**,**b**), the two Possible distributions, modulo a $2\pi/3$ rotation around the vertex, of the hole-edges when placing an octagon on a trivalent vertex such as a tetrahedron.

### 3.1. P-Cage Labeling

To label the p-cages, we follow the same notation as in [7] with a couple of differences. Each label is made of three parts: '$SYM\_Pp\_q_i$' where '$SYM$' is a label identifying the hole-polyhedron from which the p-cage is built as shown in Table 1. '$p$' is the number of edges for the p-cage faces and '$q_i$' the values for the labels a, b, c, d, and e shown on Figures 3–5. As an example, Pte_P9_2_1_3 is the p-cage made out of nonagons built from the tetrahedron with a = 2, b = 1, and c = 3. Notice that as in [7] we have excluded the truncated cuboctahedron and the truncated icosidodecahedron because the equivalence between vertices requires a reflection which we do now allow here.

**Table 1.** Symbols for convex uniform solids.

| Solid | SYM | Solid | SYM |
|---|---|---|---|
| Triangular Prism | tp | Tetrahedron | Pte |
| Square Prism (cube) | sp | Octahedron | Poc |
| Pentagonal Prism | pp | Dodecahedron | Pdo |
| Hexagonal Prism | hp | Icosahedron | Pic |
| Heptagonal Prism | 7p | Truncated Cube | Atc |
| Octagonal Prism | 8p | Truncated Tetrahedron | Att |
| Nonagonal Prism | 9p | Truncated Octahedron | Ato |
| Decagonal Prism | 10p | Truncated Dodecahedron | Atd |
| Triangular Antiprism | ta | Truncated Icosahedron | Ati |
| Square Antiprism | sa | Snub Cube | Asc |
| Pentagonal Antiprism | pa | Snub Dodecahedron | Asd |
| Hexagonal Antiprism | ha | Cuboctahedron | Aco |
| Heptagonal Antiprism | 7a | Rhombicuboctahedron | Arco |
| Octagonal Antiprism | 8a | Rhombicosidodecahedron | Arcd |
| Nonagonal Antiprism | 9a | Icosidodecahedron | Aid |
| Decagonal Antiprism | 10a | | |

The difference compared to [7] is that we have decided to use the label sp instead of Pcu for the cube. Moreover, the p-cages labeled Poc1 in [7] are identical to the triangular antiprism and we have kept the symbol Poc2 for the octahedron specific p-cages to avoid confusion.

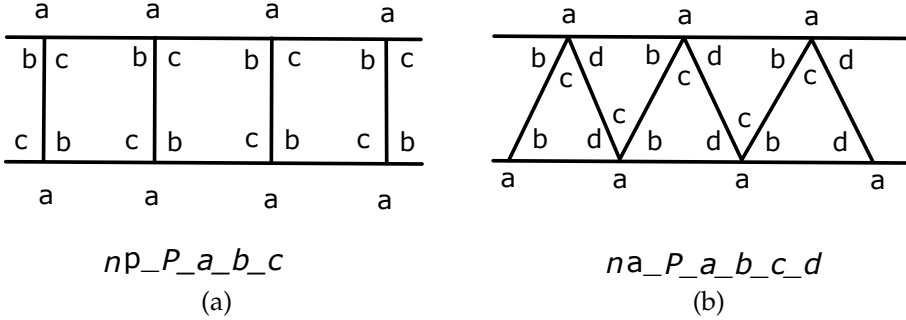

np_P_a_b_c

(a)

na_P_a_b_c_d

(b)

**Figure 3.** Distribution of the hole-edges on the (**a**) prism and (**b**) antiprism. *n* corresponds to the number of edges of the base of the prism.

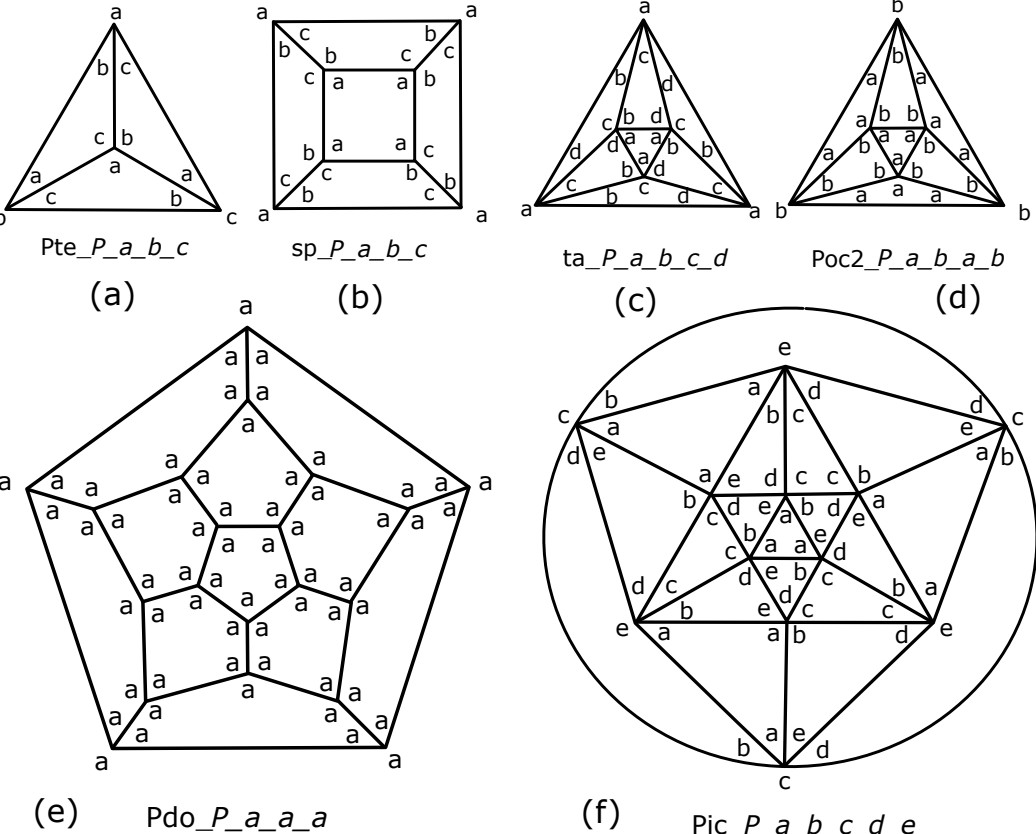

Pte_P_a_b_c

(a)

sp_P_a_b_c

(b)

ta_P_a_b_c_d

(c)

Poc2_P_a_b_a_b

(d)

(e)  Pdo_P_a_a_a

(f)  Pic_P_a_b_c_d_e

**Figure 4.** Distribution of the hole-edges on the Platonic solids: (**a**) the tetrahedron, (**b**) the cube (the square prism), (**c**) the octahedron (as a triangular antiprism (**d**)) but also a specific distribution), (**e**) the dodecahedron, and (**f**) the icosahedron.

### 3.2. Nonequivalent Hole-Edge Distribution

The distribution of hole-edges on the hole-polyhedron was done graphically in [7] for each regular solid but it can also be done using group theory. We consider one vertex of the Cayley graphs and label each face corner around a vertex with a $q_i$. We then consider each transitive subgroup of the symmetry group of the Cayley graph, as a permutation group of the vertices, and we apply these subgroups to map the initial vertex, together with the labels $q_i$, onto all the other vertices. If a $q_i$ is mapped onto a $q_j$ for $i \neq j$, then it implies that $q_i = q_j$. Such label clashes impose some constraints on the values of the $q_i$. We then obtain one mapping of the hole-edges for each transitive subgroup. The detailed construction is described in the Supplementary Materials and the result is summarized in the Figures 3–5 where, for compatibility with [7], the $q_i$ have been labeled a, b, c, d, e.

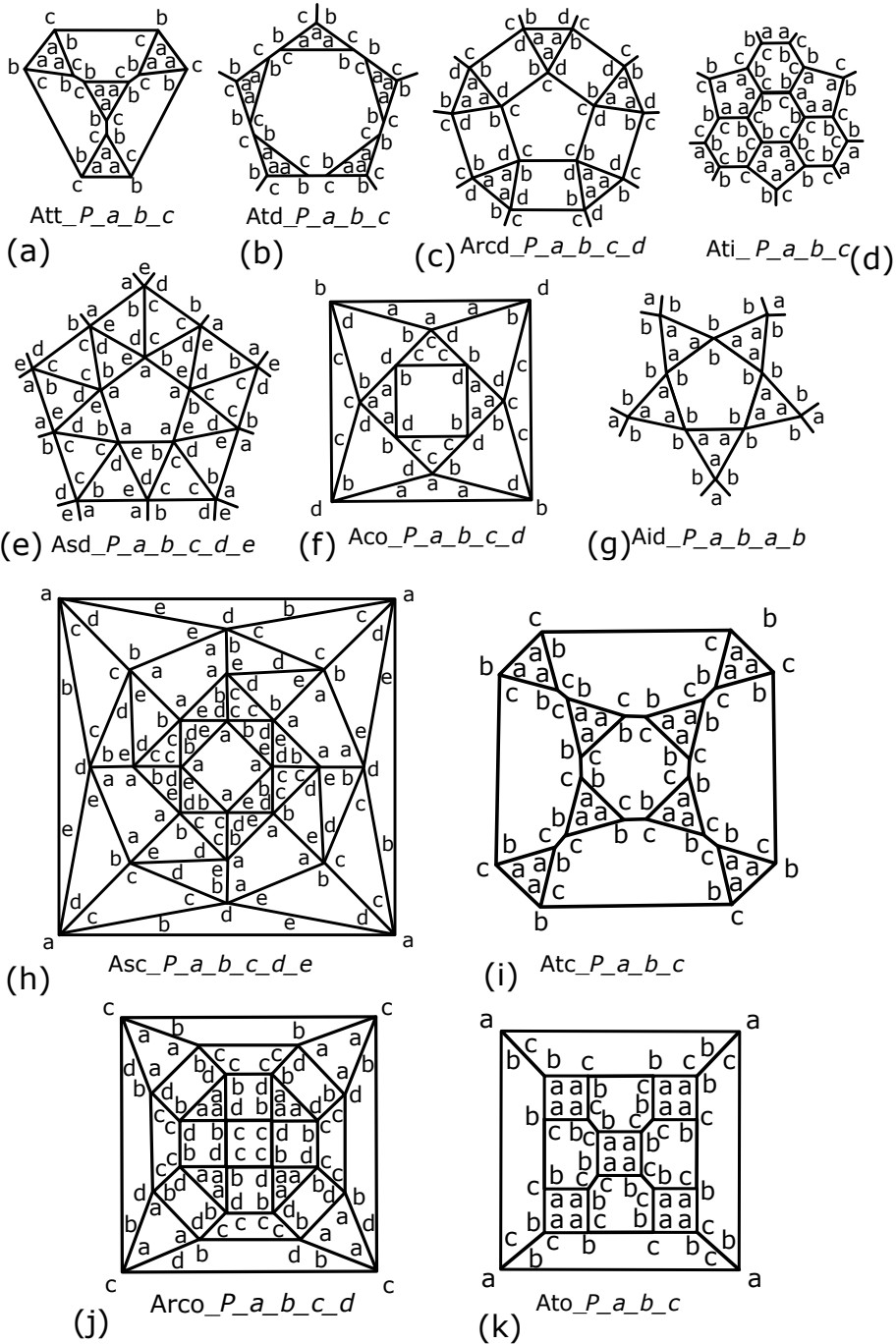

**Figure 5.** Distribution of the hole-edges on the Archimedean solids. (**a**) the truncated cube, (**b**) the truncated dodecahedron, (**c**) the rhombicosidodecahedron, (**d**) the truncated icosahedron, (**e**) the snub dodecahedron, (**f**) the cubocatahedron, (**g**) the icosidodecahedron, (**h**) the snub cube, (**i**) the truncated cube, (**j**) the rhombicuboctahedron, and (**k**) the truncated octahedron.

Because the planar graphs we are considering are those of regular solids, their symmetry groups are well known and fall into two classes [36,39,40]:

1. The dihedral groups $D_n$ generated by a rotation with an angle $2\pi/n$ around an axis, and one by an angle $\pi$ around an axis orthogonal to the first one. This is the full symmetry of the prism and antiprism.

2. The tetrahedral, octahedral, and icosahedral symmetry groups, of the Platonic and Archimedean polyhedra.

A brute force method for finding subgroups of finite groups, while in principle a finite algorithm, is too slow for practical purposes. Luckily, faster algorithms and tables for maximal subgroups are known, and have been implemented in the computer algebra software GAP; see [41] (and references therein). In what follows, we use the lists of subgroups obtained using GAP. For the tetrahedron, we have also obtained the list of subgroups by listing them in the order of their number of elements, taking into account that this must be a divisor of the order of the full group [39,40]. The process is sped up by finding first the Sylow-subgroups [39]. Alternatively, information on subgroups of finite groups is contained in [42]. As stated above, we did not consider the truncated cuboctahedron and the truncated icosidodecahedron because their symmetry group is not transitive.

The results are summarized in Tables 2 and 3. More details are provided in the Supplementary Materials.

From Table 2, we see that the prisms, the anti-prisms, the tetrahedron, the cube, and the icosahedron have one transitive subgroup which does not lead to any constraints on the labeling of the hole-edges. This implies that there is a single labeling without constraints between the $q_i$, as illustrated in Figures 3 and 4. The octahedron, which is a triangular anti-prism, has two transitive subgroups. The first one is $S_3 = D_3$, which can be applied in four different ways and which is the symmetry of the triangular anti-prism as expected. The second transitive subgroup is specific to the extra symmetry of the octahedron and it imposes the constraint $q_1 = q_3$ and $q_2 = q_4$ as illustrated in Figure 4 (where $c = a$ and $d = b$). The dodecahedron does not have any transitive subgroups and the full group symmetry imposes that the $q_i$ are all identical.

From Table 3, we see that the only Archimedean solid which has a transitive subgroup is the cuboctahedron but it does not lead to any constraints between the $q_i$. The other solids do not have any transitive subgroups and their symmetries do not induce constraints between the $q_i$ except for the icosidodecahedron where $q_1 = q_3$ and $q_2 = q_4$.

Now that we have derived all the possible connectivities between the faces of the p-cages we can construct p-cages and minimize their deformation.

**Table 2.** Symmetry groups and their transitive subgroups for p-cages with Platonic hole-polyhedra.

| Hole-Polyhedron | Symmetry (Group) | Order | Transitive (Subgroups) | Order | Label Clashes |
|---|---|---|---|---|---|
| prism | $D_n$ | $2n$ | — | | — |
| anti-prism | $D_n$ | $2n$ | — | | — |
| tetrahedron | $A_4$ | 12 | $C_2 \times C_2$ | 4 | — |
| | | | $A_4$ | 12 | all |
| cube | $S_4$ | 24 | $D_8$ | 16 | — |
| | | | $S_4$ | 24 | all |
| octahedron | $S_4$ | 24 | $4\,S_3 \equiv 4\,D_3$ | 6 | — |
| | | | $A_4$ | 12 | two pairs |
| | | | $S_4$ | 24 | all |
| dodecahedron | $A_5$ | 60 | — | | all |
| icosahedron | $A_5$ | 60 | $A_4$ | 12 | — |
| | | | $A_5$ | 60 | all |

**Table 3.** Symmetry groups and their transitive subgroups for p-cages with Archimedean hole-polyhedra. $G$ denotes the full symmetry group of the hole-polyhedron and $H$ the transitive subgroup considered.

| Hole Polyhedron | Symmetry Type | $G$ | $\|G\|$ | $H \subseteq G$ | $\|H\|$ | Label cl's. |
|---|---|---|---|---|---|---|
| truncated tetrahedron | tetrahedron | $A_4$ | 12 | — | | |
| truncated cube | cube | $S_4$ | 24 | — | | |
| truncated octahedron | octahedron | $S_4$ | 24 | — | | |
| truncated dodecahedron | dodecahedron | $A_5$ | 60 | — | | |
| truncated icosahedron | icosahedron | $A_5$ | 60 | — | | |
| cuboctahedron | cube | $S_4$ | 24 | $A_4$ | 12 | — |
| | | | | $S_4$ | 24 | two pairs |
| rhombicuboctahedron | cube | $S_4$ | 24 | | | |
| snub cube | cube | $S_4$ | 24 | | | |
| icosidodecahedron | dodecahedron | $A_5$ | 60 | | | two-pairs |
| rhombicosidodecahedron | dodecahedron | $A_5$ | 60 | | | |
| snub dodecahedron | dodecahedron | $A_5$ | 60 | | | |

## 4. Near-Miss Convex Symmetric P-Cages

For a p-cage to be regular, all the faces, all the edges, and all the angles of the polygonal faces must be identical. For a $P$-gon, this means that all the edges must have the same length $L$ and the same angle $\pi(1 - 2/P)$. Near-miss p-cages are p-cages where the faces are not regular polygons, but close to being regular. Irregular faces will have edge lengths and angles slightly different from the values of a regular one.

To evaluate the level of regularity of the p-cage we first determine the distance $d_i$ between vertices $i$ and $i + 1$ as well as the angle $\alpha_i$ between the segments $(i - 1, i)$ and $(i, i + 1)$. The function we must minimize is then:

$$E = \frac{1}{P} \sum_i \left[ c_l \left( \frac{d_i - L}{L} \right)^2 + c_a \left( \frac{\alpha_i - \pi(1 - \frac{2}{P})}{\pi(1 - \frac{2}{P})} \right)^2 \right] + c_c \, E_{\text{conv}} \tag{1}$$

where $c_l$, $c_a$, and $c_c$ are three weight factors. $E_{\text{conv}}$, given explicitly by (3), is 0 unless the polygon defined by the $n_i$ is concave. It is used in the simulated annealing to enforce convexity of the faces by taking a large value of $c_c$. We divide the sum by $P$ to approximately set the same energy scale for each $P$. This makes the parametrization of the optimizing algorithm easier.

To characterize the face, with normal vector $m_f$, we define $n_i$ as its vertices, ordered anticlockwise. Then, to measure the angle $\alpha_i$ and edge length $d_i$, we define $v_i = n_i - n_{i-1}$, evaluate $v_i \times v_{i+1}$ and

$$\text{if } (v_i \times v_{i+1}) \cdot m_f \geq 0 \quad : \quad \alpha_i = \pi - \text{acos}(\frac{(v_i \cdot v_{i+1})}{|v_i||v_{i+1}|}), \quad d_i = |v_i|$$

$$\text{if } (v_i \times v_{i+1}) \cdot m_f < 0 \quad : \quad \alpha_i = \pi + \text{acos}(\frac{(v_i \cdot v_{i+1})}{|v_i||v_{i+1}|}), \quad d_i = |v_i|. \tag{2}$$

Note that $\alpha_i$ in (2) corresponds to the angle inside the face which is larger than $\pi$ if the face is not convex. If $m_f$ is the vector normal to the face and if the $n_i$ are running

anticlockwise when seeing the face in the direction of $n_f$, then, using the Heaviside function $H(x)$:

$$E_{\text{conv}} = \frac{1}{P} \sum_i \left[ H\left( (v_i \times v_{i+1}) \cdot m_f \right) \right]. \tag{3}$$

We then define the length and angle deformations as follows:

- Length : $\Delta_l = \max_i(|\frac{d_i - L}{L}|)$
- Angle : $\Delta_a = \max_i(|\frac{\alpha_i - \pi(1 - \frac{2}{p})}{\pi(1 - \frac{2}{p})}|$

In most cases near-miss p-cages can be deformed smoothly, changing the edge lengths as well as the angles and as a result both $\Delta_l$ and $\Delta_a$. Identifying near-miss p-cages for a given connectivity (fixed hole-polyhedra, $P$ and $q_i$) consists in finding the geometry which minimizes $\Delta_l$ and $\Delta_a$. This can be done by minimizing the function (1) over the coordinates of the vertices. As in [7], we do this using a simulated annealing algorithm, for a range of values of $c_l$ and $c_a$ satisfying the constraint $c_l + c_a = 2$. After removing from the obtained p-cages those with crossing faces, we have selected the configuration with the smallest deformation, i.e., those for which the maximum value of $\Delta_l$ and $\Delta_a$ is the smallest.

The regular convex p-cages ($\Delta_l = \Delta_a = 0$) were derived analytically in [7].

## 5. Optimization of the P-Cage Coordinates

To build the symmetric p-cage, we consider one face, which we refer to as the reference face, as well as the plane that it spans. The normal vector to that plane can have a range of orientations, and we apply the symmetry of the underlying solid to that plane to generate the planes spanned by the adjacent faces. We then determine the lines of intersection between these neighbor faces and place the shared vertices onto these lines. Notice that as the p-cage faces have holes, the actual symmetry of the p-cages allows for some additional deformation as will be described below. The vertices which are not shared can then be placed on the reference face plane. One can then determine the most general analytic expression for the coordinates of the vertices of the reference face for each of the hole-polyhedra graphs. These expressions depends on a number of parameters which can then be optimized to obtain the p-cage minimizing the deformation energy (1). We did this for 200 values of the weight parameters $c_l$ and $c_a$ such that they satisfy the constraint $c_l + c_a = 2$. Once the coordinates of the reference face have been obtained, one can obtain the coordinates of the full p-cage vertices by applying the symmetry group of the p-cage.

One of the problems we will have to solve is to find the intersection between two planes defined by:

$$\mathcal{P}_1(t_1, t_2) = V + t_1 v_1 + t_2 v_2, \qquad \mathcal{P}_2(s_1, s_2) = W + s_1 w_1 + s_2 w_2, \tag{4}$$

where the $t_1, t_2, s_1, s_2$ are parameters, $V$ and $W$ are arbitrary vectors, and where the plane basis vectors $v_1$ and $v_2$ can be assumed to be orthonormal, and similarly for $w_1$ and $w_2$.

First we define the normal vectors, $p$ and $q$, to the planes as well as the vector $u$ parallel to the plane intersection:

$$p = v_1 \times v_2, \qquad q = w_1 \times w_2, \qquad u = q \times p. \tag{5}$$

Next, we have to find a specific point on the intersecting line and we choose the one that is perpendicular to $u$:

$$U = V + t_1 v_1 + t_2 v_2 = W + s_1 w_1 + s_2 w_2 \tag{6}$$

and multiplying (6) by $u$ leads to a relation between $t_1$ and $t_2$ as well as $s_1$ and $s_2$. Then multiplying (6) by $q$ one obtains an expression for $t_1$ which when inserted back into (6) gives:

$$U \;=\; V + \frac{(q \cdot (W - V))(u \cdot v_2) + (u \cdot V)(q \cdot v_2)}{(q \cdot v_1)(u \cdot v_2) - (u \cdot v_1)(q \cdot v_2)} \left( v_1 - \frac{(u \cdot v_1)}{(u \cdot v_2)} v_2 \right) - \frac{(u \cdot V)}{(u \cdot v_2)} v_2. \quad (7)$$

We are now ready to construct each family of p-cages one by one by considering their specific symmetries.

### 5.1. P-Cages

We now consider each generating regular solid in turn and generate the planes making the p-cage using the specific symmetry of the solid and determine the line of intersection where the shared segment lies. Note that the length of $V$ in (4) is arbitrary as it will be adjusted to set the scale of the p-cage so that the edge lengths are as close as possible to the target length (chosen arbitrarily to be 1). For many p-cages, there will also be one or two angles corresponding to deformations of the dual of the generating regular solid.

Some of the coordinates of the vertices on the shared segments will be of the form:

$$n_i \;=\; U + t_i u \quad (8)$$

where $U$ and $u$ are specific to each segments. The remaining shared vertices will be of the form:

$$n_i \;=\; R\, n_j \quad (9)$$

where $R$ is a rotation symmetry. The details will vary according to the generating regular solid.

The coordinates of the vertices which are not part of a shared edge will be of the form:

$$\alpha_i \;=\; V + t_{\alpha_i} v_1 + s_{\alpha_i} v_2, \qquad i = 1 \dots q_\alpha - 1 \quad (10)$$

where $q_\alpha$ stands for $a$, $b$, $c$, $d$, or $e$, as appropriate. Once this is done, the actual position of the vertices will be determined by minimizing (1), the optimizing variables being the length of $V$, some angles describing the relative orientation between some of the faces as well as the position of all the vertices on the face plane.

### 5.2. Tent

We shall see that a common substructure of the dual of the generating regular solid is a pyramid. When the faces of the p-cage are placed on a pyramid, they each have three shared segments, two with the adjacent faces of the pyramid and one at the base. We call such a structure a tent. The shared segment at the base does not need to be centered and, as we shall see, does not need to be parallel to the base either. This is because two adjacent pyramids can sometimes be rotated around their main symmetry axis. As we shall see, for the Archimedean based p-cages, the tents will not have a common symmetry axis.

The origin of the coordinate system for the reference face is chosen as the center of the base of the pyramid (Figure 6). The $x$ axis runs parallel to the edge of the base of the pyramid while the $y$ axis is parallel to the pyramid base and perpendicular to the $x$ axis. The inclination angle of the pyramid face is $\theta$.

The edges joining $n_1$ and $n_2$ as well as $n_3$ and $n_4$ on Figure 6 are shared edges. So is the edge joining $n_5$ and $n_6$. The vertices that are not shared must also belong to the plane of the face.

If the base of the pyramid is an $N$-gon, $n_1$ and $n_4$ are related by a $2\pi/N$ rotation around the vertical symmetry axis, $z$, of the pyramid. So are $n_2$ and $n_3$. $n_5$ and $n_6$ will be related to each other by a rotation which depends on how adjacent tents are placed with respect to each other.

In what follows, we shall use the notation $R_v(\theta)$ to denote a $\theta$ rotation around the vector $v$. We also use $T_v$ to denote a reflection with respect to the $v$ axis. If $v$ is $x$, $y$, or $z$ it then corresponds to a rotation, or a reflection, around the corresponding axis.

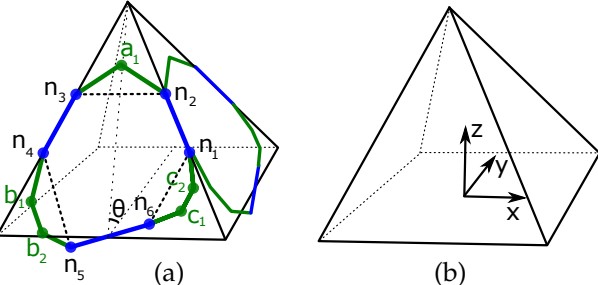

**Figure 6.** (**a**) Parametrization of a tent. The edges joining $n_1$ to $n_2$, $n_3$ to $n_4$, and $n_5$ to $n_6$, are shared edges. The edges adjacent to $a_i$, $b_i$, and $c_i$ are hole-edges. $\theta$ is the inclination of the face with respect to the pyramid base. (**b**) Coordinate frame.

*5.3. Specific Cases*

In what follows, $r_f$ and $r_o$ refer respectively to the inner and outer radius of the face of a regular solid, while $r_i$ refers to the inner radius of the regular solid (the radius of the smallest sphere containing it).

Prism P-Cages

The dual of a prism consists of two identical pyramids joined at their base. The faces of the pyramids can have any angle with the base. For a pyramid with an N-gon base, the symmetries are rotations of $\gamma = 2\pi/N$ around the main axis $z$ of the pyramid. (See Figure 6). To map the two pyramids into each other we must perform a $\pi$ rotation around the $y$ axis joining the center of the base and the center of one of the base edge. Alternatively we can perform a $\pi$ rotation around the $z$ axis followed by a $\pi$ rotation around the $x$ axis.

Because the faces of a p-cage do not cover the full face of the pyramid, we can also rotate the pyramids around the $z$ axis by an angle $\psi$. So to map the two tents together, we must perform a $\pi + \psi$ rotation around the $z$ axis followed by a $\pi$ rotation around the $x$ axis.

To describe the plane of the reference face we define:

$$V = S(0, -\sin(\theta), \cos(\theta)), \qquad v_1 = ([1,0,0]), \qquad v_2 = \frac{V \times v_1}{|V \times v_1|}, \qquad (11)$$

where $S$ is a scaling parameter. As a result, the adjacent face belonging to the same pyramid is given by:

$$W = R_z(\frac{2\pi}{N})\, V, \qquad w_1 = R_z(\frac{2\pi}{N})\, v_1, \qquad w_2 = R_z(\frac{2\pi}{N})\, v_2$$

$$(12)$$

and using the corresponding $U$ and $u$, (7):

$$n_1 = U - t_1\, u, \quad n_2 = U + t_2\, u, \quad n_3 = R_z(\frac{-2\pi}{N})\, n_2, \quad n_4 = R_z(\frac{-2\pi}{N})\, n_1. \qquad (13)$$

For the vertices at the base of the tent:

$$W = R_x(\pi)R_z(\pi + \psi)\, V, \quad w_1 = R_x(\pi)R_z(\pi + \psi)\, v_1, \quad w_2 = R_x(\pi)R_z(\pi + \psi)\, v_2 \qquad (14)$$

and using the corresponding $U$ and $u$:

$$n_5 = U - t_5\, u, \qquad n_6 = R_x(\pi)R_z\, n_5. \qquad (15)$$

The optimizing variables are $S$, $\theta$, $\psi$, $t_1$, $t_2$, $t_5$, Equations (11), (13), (14), and (15), as well as the coordinates, in the face plane, of the non shared vertices.

*5.4. Regular Flat Prisms P-Cages*

A degenerate case of p-cages corresponds to flat p-cages, i.e., $\theta = 0$. This implies that the $n_1 - n_2$ and $n_3 - n_4$ lines intersect each other at the center of the prism base. The angle between these two lines is $\pi - 2(q_a + 1)\pi/P$ and this must equal $2\pi/N$. In other words, for a given polygon $P$ and prism base $N$:

$$q_a = \frac{(N-2)P}{2N} - 1. \tag{16}$$

The list of regular flat prisms is given in Table 4.

**Table 4.** List of regular flat prism p-cages. See the Supplementary Materials for the full list of non-planar p-cages.

| N | 3 | 4 | 4 | 5 | 6 | 6 | 6 | 6 | 6 | 8 | 10 | 10 |
|---|---|---|---|---|---|---|---|---|---|---|----|----|
| P | 12 | 8 | 16 | 20 | 6 | 9 | 12 | 15 | 18 | 16 | 19 | 20 |
| $q_a$ | 1 | 1 | 3 | 5 | 1 | 2 | 3 | 4 | 5 | 5 | 3 | 7 |
| $q_b$ | 4 | 2 | 5 | 6 | 1 | 2 | 3 | 4 | 5 | 4 | 2 | 5 |
| $q_c$ | 4 | 2 | 5 | 6 | 1 | 2 | 3 | 4 | 5 | 4 | 2 | 5 |

## 6. Antiprisms

The dual of an antiprism can be seen as two pyramids joined at the base and rotated by an angle $\pi/N$ relative to each other, resulting in each face having four neighbors: two on the sides and two at the base.

The parametrization is similar to that of the prism p-cage and the vertices $n_1$ to $n_4$ are also given by (13).

For the first pair of shared vertices at the base:

$$\mathcal{R} = R_x(\pi)R_z(\pi(1 + \frac{1}{N}) + \psi), \qquad W = \mathcal{R}\,V, \quad w_1 = \mathcal{R}\,v_1, \quad w_2 = \mathcal{R}\,v_2 \tag{17}$$

and using the corresponding $U$ and $u$:

$$n_5 = U - t_5\,u, \qquad n_6 = R_x(\pi)R_z\,n_5. \tag{18}$$

For the second pair of shared vertices:

$$\mathcal{R} = R_x(\pi)R_z(\pi(1 - \frac{1}{N}) + \psi), \qquad W = \mathcal{R}\,V, \quad w_1 = \mathcal{R}\,v_1, \quad w_2 = \mathcal{R}\,v_2 \tag{19}$$

and using the corresponding $U$ and $u$:

$$n_7 = U - t_7\,u, \qquad n_8 = R_x(\pi)R_z\,n_7. \tag{20}$$

The optimizing variables are $S$, $\theta$, $\psi$, $t_1$, $t_2$, $t_5$, $t_7$, Equations (11), (13), (17), (18), and (20), as well as the coordinates, in the face plane, of the non shared vertices.

## 7. Platonic Solids

*7.1. Tetrahedron*

The generating regular solid of the tetrahedron p-cage is a tetrahedron. As a result, the face of the p-cage must be placed on the faces of the dual tetrahedron but it does not need to be regular and can be elongated in such a way that the line joining the center of two pairs of opposite edges is stretched.

The coordinates of the four vertices of the tetrahedron (Figure 7) are:

$$e_1 = (1, 1, -a), \quad e_4 = (-1, -1, -a), \quad e_3 = (1, -1, a), \quad e_4 = (-1, 1, a). \tag{21}$$

where $a$ is a stretching parameter. We choose the first face as spanned by $e_1, e_2, e_3$ and the center of that face is $V_0 = (e_1 + e_2 + e_3)/3$. We then take:

$$v_1 = \left( \frac{1}{\sqrt{2}}, \quad \frac{1}{\sqrt{2}}, 0 \right), \quad v_2 = e_3 - (e_1 + e_2)/2. \tag{22}$$

Noticing that $V$ is not orthogonal to $v_2$, we must take as the displacement vector for the face plane:

$$V = S(v_1 \times v_1)((v_1 \times v_1) \cdot V_0), \tag{23}$$

where $S$ is a scaling parameter. Defining:

$$\mathcal{R}_1 = R_{v_1}(\pi)R_z(\pi/2), \qquad \mathcal{R}_2 = R_{v_1}(\pi)R_z(-\pi/2), \qquad \mathcal{R}_3 = R_z(\pi) \tag{24}$$

the vector describing the adjacent faces are:

$$W^i = \mathcal{R}_i V, \qquad w_1^i = \mathcal{R}_i v_1, \qquad w_2^i = \mathcal{R}_i v_2 \tag{25}$$

and the corresponding $U^i$ and $u^i$:

$$n_{2i-1} = U^i + t_{2i-1} u^i, \quad n_{2i} = \mathcal{R}_i^{-1} n_{2i-1}, \qquad i = 1, 2, 3. \tag{26}$$

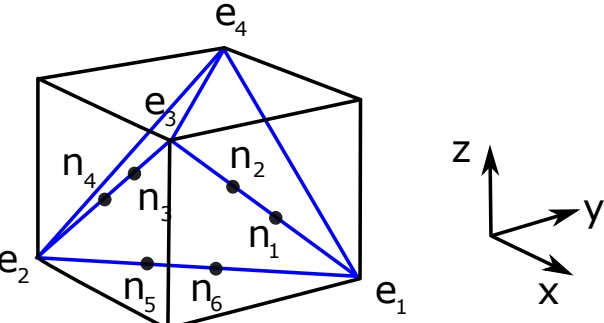

**Figure 7.** Tetrahedron inside a cube.

The optimizing variables are $S$, $a$, $t_1$, $t_3$, $t_5$, Equations (21), (23), and (26), as well as the coordinates, in the face plane, of the non shared vertices.

### 7.2. Octahedron

The dual of the octahedron is the cube, so the face of the octahedron derived p-cages will be inscribed on the faces of a cube.

The cube can be stretched in one direction so that two of the faces remain square but the other four become identical rectangles, the symmetry is that of a square antiprism, referred to as Poc1 in [7].

When the cube is not stretched, the symmetry is larger and the shared vertices ab are facing shared vertices ba. This is referred to as Poc2, as in [7].

There are two ways to map the vertices on the face of the cube. The first possibility is to pinch the cube, making the faces diamond shape, but for the edges to meet correctly they must be centered on the edges of the diamond. This is nothing but the triangular antiprism with $a = c$ and $b = d$.

In the second one, the squares are not deformed and the shared edges are not centered on the edges of the squares. To preserve the symmetry of the hole distribution, the cube must remain undeformed. To describe the plane of the reference face we define:

$$V = (0, 0, S) \quad v_1 = (1, 0, 0), \quad v_2 = (0, 1, 0) \tag{27}$$

and the symmetries are $\pm \pi/2$ rotations around the $x$ and the $y$ axis:

$$R_1 = R_y(\pi/2), \quad R_2 = R_x(\pi/2). \tag{28}$$

Next, the adjacent faces are described by the vectors:

$$W^i = \mathcal{R}_i V, \qquad w_1^i = \mathcal{R}_i v_1, \qquad w_2^i = \mathcal{R}_i v_2, \quad i = 1, 2 \tag{29}$$

together with the corresponding $U_i$ and $u_i$.

$$
\begin{aligned}
n_1 &= U^1 + t_1 u^1, & n_2 &= U^1 + t_2 u^1, & n_3 &= U^2 - t_2 u^2, \\
n_4 &= U^2 - t_1 u^2, & n_{4+i} &= R_z(\pi) n_i, \quad i = 1, 4.
\end{aligned}
\tag{30}
$$

The optimizing variables are $S, t_1, t_2$, Equations (27) and (30), as well as the coordinates, in the face plane, of the non shared vertices.

### 7.3. Cube

The p-cages derived from the cube are the same as those derived from the square prism.

### 7.4. Dodecahedron

The p-cages derived from the dodecahedron must have identical holes, so the symmetry is very constrained. The faces must be placed on an icosahedron.

As the inner radius of an icosahedron of edge length 1 is $r_i = \phi_g^2/\sqrt{12}$ where $\phi_g = (1 + \sqrt{5})/2$, and with $r_o = 1/(2 \sin(2\pi/3))$ we take:

$$V = r_i(0, 0, S) \quad v_1 = (1, 0, 0), \quad v_2 = (0, 1, 0). \tag{31}$$

where $S$ is a scaling factor. We place the three vertices at:

$$f_1 = V + (0, S r_o, 0), \quad f_{i+1} = R_z(i \frac{2\pi}{3}) f_1 \quad i = 1, 2. \tag{32}$$

The midpoint between the first and last vertex is:

$$a = \frac{f_1 + f_3}{2} \tag{33}$$

and the adjacent faces are described by the vectors:

$$W = R_{f_1}(\frac{2\pi}{5}) V, \qquad w_1 = R_{f_1}(\frac{2\pi}{5}) v_1, \qquad w_2 = R_{f_1}(\frac{2\pi}{5}) v_2. \tag{34}$$

and the line intersecting the reference face is described by the corresponding $U$ and $u$. From this we obtain:

$$
\begin{aligned}
n_1 &= U + tu, & n_2 &= R_a(\pi), n_1, \\
n_{2i+1} &= R_z(\frac{2\pi i}{3}), n_1, & n_{2i+2} &= R_z(\frac{2\pi i}{3}), n_2, \quad i = 1, 2.
\end{aligned}
\tag{35}
$$

The optimizing variables are $S, t$, Equations (31) and (35), as well as the coordinates, in the face plane, of the non shared vertices.

### 7.5. Icosahedron

The p-cages derived from the icosahedron consist in placing the faces on a dodecahedron. As the inner radius of a dodecahedron of edge length 1 is $r_i = \phi_g^2/(2\sqrt{3 - \phi_g})$, and $r_o = 1/(2\sin(\pi/5))$, we take:

$$V = r_i(0, 0, S), \quad v_1 = (1, 0, 0), \quad v_2 = (0, 1, 0). \tag{36}$$

where $S$ is a scaling factor. We place the five vertices of the dodecahedron at:

$$f_1 = V + (0, S\, r_o, 0), \quad f_{i+1} = R_z(i\frac{2\pi}{5})\, f_1, \quad i = 1, 2, 3, 4. \tag{37}$$

Next, the adjacent faces are described by the vectors:

$$W^i = R_{f_i}(\frac{2\pi}{3})V, \quad w_1^i = R_{f_i}(\frac{2\pi}{3})v_1, \quad w_2^i = R_{f_i}(\frac{2\pi}{3})v_2, \quad i = 1, .., 5. \tag{38}$$

together with the corresponding $U^i$ and $u^i$. Defining $T_x$ as a reflection around the $x$ axis, (Figure 8):

$$n_1 = U^1 + t_1 u^1, \qquad n_2 = U^1 + t_2 u^1, \qquad n_3 = T_x\, n_2, \quad n_4 = T_x\, n_1,$$

$$n_5 = U^3 + t_5 u^3, \qquad n_6 = U^3 + t_6 u^1, \qquad n_7 = R_z(\frac{4\pi}{2})\, T_x\, R_z^t(\frac{4\pi}{2})\, n_2,$$

$$n_8 = R_z(\frac{4\pi}{2})\, T_x\, R_z^t(\frac{4\pi}{2})\, n_1, \qquad n_9 = U^5 + t_9 u^5, \qquad n_{10} = R_{(f_4 + f_5)}(\pi)\, n_9. \tag{39}$$

The optimizing variables are $S$, $t_1$, $t_2$, $t_5$, $t_6$, $t_9$, Equations (36) and (39), as well as the coordinates, in the face plane, of the non shared vertices.

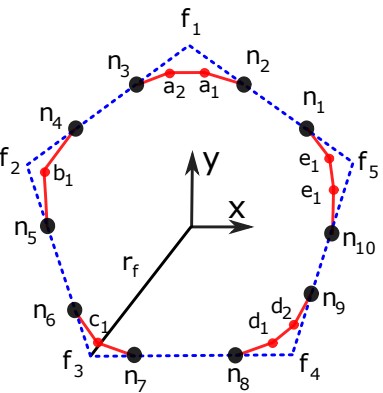

**Figure 8.** Face of an icosahedron p-cage.

## 8. Archimedean Solids

### 8.1. Truncated Platonic Solids

The p-cages generated from a truncated Platonic solid can be obtained by placing a tent on the faces of the dual solid, referred to as the underlying solid. For example, p-cages are obtained from the truncated cube by placing a triangular tent on the faces of an octahedron (Figure 9).

- Truncated tetrahedron: triangular tent on the faces of a tetrahedron.
- Truncated cube: triangular tent on the faces of an octahedron.
- Truncated octahedron: square tent on the faces of a cube.
- Truncated dodecahedron: triangular tent on the faces of a icosahedron.
- Truncated icosahedron: pentagonal tent on the faces of a dodecahedron.

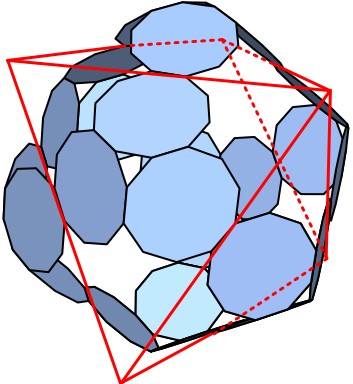

**Figure 9.** Triangular tents placed on the faces of an octahedron for the p-cage constructed from the truncated cube.

The side vertices, $n_1 - n_4$ are obtained exactly like for the prism (Figure 6). The bottom vertices, $n_5$ and $n_6$ are obtained by finding the intersection between the plane of the first face and the plane obtained by a $\pi$ rotation of that face around the vector $V_0 = (r_f \sin(\psi), -r_f \cos(\psi), r_i)$ where $r_f$ and $r_i$ are properties, listed in Table 5, of the Platonic solid on which the p-cage faces are placed.

The vertices $n_5$ and $n_6$ are then placed symmetrically around the point of intersection between the line span by $V_0$ and the intersection between the planes of the two faces.

The normal to the faces is $V$ given by (11) and for the cage to be convex the angle between $V$ and the normal to the Platonic solid face on which the tent is built, $\theta = \arccos((V \cdot \hat{e}_x))$, must be smaller than $\pi - \phi_d$ where $\phi_d$ is the dihedral angle of the underlying Platonic solid.

**Table 5.** Inner radius, $r_i$, inner face radius, $r_f$, and dihedral angle of Platonic solid of edge length 1. $\phi_g = (1 + \sqrt{5})/2$.

| Solid | $r_i$ | $r_f$ | $\phi_d$ |
|---|---|---|---|
| Tetrahedron | $1/\sqrt{24}$ | $1/(2\tan(\frac{\pi}{3}))$ | $2\arctan(1/\sqrt{2})$ |
| Cube | $1/2$ | $1/2$ | $\pi/2$ |
| Octahedron | $1/\sqrt{6}$ | $1/(2\tan(\frac{\pi}{3}))$ | $2\arctan(\sqrt{2})$ |
| Dodecahedron | $\phi_g^2/(2\sqrt{3-\phi_g})$ | $1/(2\tan(\frac{\pi}{5}))$ | $2\arctan(\phi_g)$ |
| Icosahedron | $\phi_g^2/\sqrt{12}$ | $1/(2\tan(\frac{\pi}{3}))$ | $2\arctan(\phi_g^2)$ |

*8.2. Cuboctahedron*

The dual of the cuboctahedron is a rhombic dodecahedron which is made out of twelve identical rhombi. This can also be thought of as four triangular pyramids placed on the faces of a tetrahedron. The faces of the pyramids are located near the corner of the triangles so as to be adjacent to two faces of two other pyramids. This can be done in two different ways, around the triple 'a' vertex or around the the triple 'c' one. In both cases, the quadrivalent holes are mapped on the edges of the tetrahedron (Figure 10).

We take the following coordinates for the tetrahedron, $f_1 = (1/2, 1/2\sqrt{3}, 1/2\sqrt{6})$, $f_2 = (-1/2, 1/2\sqrt{3}, 1/2\sqrt{6})$, $f_3 = (0, -1/\sqrt{3}, 1/2\sqrt{6})$, and $f_4 = (0, 0, -\sqrt{3/8})$, which correspond to an upside down pyramid.

We choose the following vectors:

$$V = S(0, -\sin(\theta), \cos(\theta)), \qquad v_1 = (1, 0, 0), \qquad v_2 = \frac{V \times v_1}{|V \times v_1|}, \qquad (40)$$

where $S$ is a scaling parameter, to describe the plane of the reference face. The adjacent face belonging to the same pyramid is then given by :

$$W^1 = R_z(\frac{2\pi}{3})\,V, \qquad w_1^1 = R_z(\frac{2\pi}{3})\,v_1, \qquad w_2^1 = R_z(\frac{2\pi}{3})\,v_2 \tag{41}$$

and using the corresponding $U$ and $u$:

$$n_1^1 = U + t_1\,u, \quad n_2 = U + t_2\,u, \quad n_3^1 = R_z(\frac{-2\pi}{3})\,n_2 \quad n_4^1 = R_z(\frac{-2\pi}{3})\,n_1. \tag{42}$$

For the remaining four vertices, we must consider the axis $f_3$ as an axis of symmetry which can also be rotated by an angle $\psi$ around the $z$ axis: $g_3 = R_z(\psi)f_3$. Then the adjacent face containing $n_7$ and $n_8$ is spanned by the vectors:

$$W^2 = R_{g_3}(\frac{2\pi}{3})\,V, \qquad w_1^2 = R_{g_3}(\frac{2\pi}{3})\,v_1, \qquad w_2^2 = R_{g_3}(\frac{2\pi}{3})\,v_2 \tag{43}$$

and using the corresponding $U$ and $u$:

$$\begin{aligned} n_7^2 &= U + t_7\,u & n_8 &= U + t_8\,u \\ n_5^2 &= R_{g_3}(\frac{-2\pi}{3})\,n_8 & n_6^2 &= R_{g_3}(\frac{-2\pi}{3})\,n_7. \end{aligned} \tag{44}$$

The optimizing variables are $S$, $t_1$, $t_2$, $t_7$, and $t_8$, Equations (40), (42), and (44), as well as the coordinates, in the face plane, of the non shared vertices.

The rhombic dodecahedron has twelve quadrilateral faces. These faces have two vertices with three neighbors and two with four. One can split the quadrivalent vertices in two by inserting an extra edge and do so to obtain the graph of the dodecahedron. This means that the regular p-cages obtained by tiling the faces of the dodecahedron such that the added edge matches a shared edge of the p-cage face corresponds to a degenerate symmetric p-cage derived from the cubocatehedron. More explicitly, `Aco_P10_1_1_1_3`, `Aco_P15_2_2_2_5`, and `Aco_P20_3_3_3_7` are degenerate cases of `Pic_P10_1_1_1_1_1`, `Pic_P15_2_2_2_2_2`, and `Pic_P20_3_3_3_3_3`, respectively .

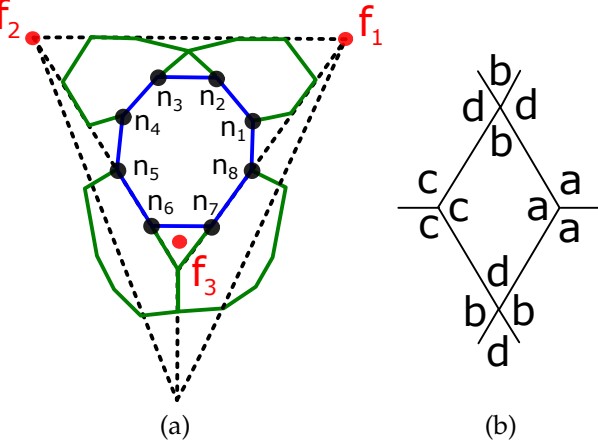

(a)  (b)

**Figure 10.** (**a**) Parametrization of the cuboctahedron p-cage face. (**b**) Distribution of hole-edges.

### 8.3. Icosidodecahedron

The dual of the icosidodecahedron is a rhombic tricontahedron which is made out of 30 identical rhombi. The p-cage is obtained by inscribing a P-gon inside that rhombus face (Figure 11).

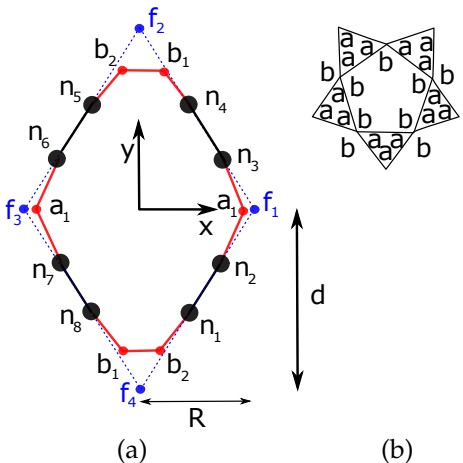

**Figure 11.** (**a**) Parametrization of the icosidodecahedron p-cage face. (**b**) Distribution of hole-edges.

The line $n_1 - n_2$ is obtained by determining the intersection of the plane containing the master face and the one obtained after a $2\pi/3$ rotation about the axis $f_1$. Then, $n_3$ and $n_4$ are obtained by performing a reflection around the $f_1 - f_3$ axis of $n_2$ and $n_1$, respectively. The vertices $n_5$, $n_6$, $n_7$, and $n_8$ are obtained by performing a reflection around the $f_2 - f_4$ axis of $n_4$, $n_3$, $n_2$, and $n_1$, respectively.

The coordinates of the rhombic tricontahedron are given by:

$$\left( \pm\phi_g^2, \pm\phi_g^2, \pm\phi_g^2 \right) \qquad \text{8 vertices} \tag{45}$$

$$\left( \pm\phi_g^3, \pm\phi_g, 0 \right) \qquad +\text{cycl.} : 12 \text{ vertices} \tag{46}$$

$$\left( \pm\phi_g^2, \pm\phi_g^3, 0 \right) \qquad +\text{cycl.} : 12 \text{ vertices} \tag{47}$$

The first 20 vertices are trivalent while the last 12 are pentavalent.

Up to a scaling factor, we can take $f_1 = (\phi_g^2, 0, \phi_d^3)$, $f_2 = (0, \phi_g, \phi_d^3)$, $f_3 = (-\phi_g^2, 0, \phi_d^3)$, and $f_4 = (0, -\phi_g, \phi_d^3)$. The center of the master face is then $V_f = (f_1 + f_3)/2 = (0, 0, \phi_g^3)$.

We take:

$$V = S\, V_f, \quad v_1 = (1, 0, 0), \quad v_2 = (0, 1, 0) \tag{48}$$

and

$$W^i = R_{f_i}(\frac{2\pi}{3})V, \quad w_1^i = R_{f_i}(\frac{2\pi}{3})v_1, \quad w_2^i = R_{f_i}(\frac{2\pi}{3})v_2, \quad i = 1, 3, \tag{49}$$

together with the corresponding $U^1$, $u^1$, $U^3$, and $u^3$.

Then:

$$
\begin{aligned}
n_1 &= U^1 + t_1 u^1, & n_2 &= U^1 + t_2 u^1, & n_3 &= T_x\, n_2, & n_4 &= T_x\, n_1, \\
n_5 &= T_y\, n_4, & n_6 &= T_y\, n_3, & n_7 &= T_y\, n_2, & n_8 &= T_y\, n_1.
\end{aligned} \tag{50}
$$

The optimizing variables are $S$, $t_1$, $t_2$, Equations (48) and (50), as well as the coordinates, in the face plane, of the non shared vertices.

*8.4. Rhombicuboctahedron*

The dual of the rhombicuboctahedron is the deltoidal icositetrahedron. It is made out of twenty-four kite faces. Three of the vertices of the faces are quadrivalent while the fourth one is trivalent. Up to an overall scale, they are at the following positions:

$$(\pm 1, 0, 0), \qquad\qquad\qquad +\text{cycl.} : 8\,\text{quadrivalent vertices} \qquad (51)$$

$$\left(0, \pm\frac{1}{\sqrt{2}}, \pm\frac{1}{\sqrt{2}}\right), \qquad\qquad +\text{cycl.} : 12\,\text{quadrivalent vertices} \qquad (52)$$

$$\left(\pm\frac{\sqrt{8}+1}{7}, \pm\frac{\sqrt{8}+1}{7}, \pm\frac{\sqrt{8}+1}{7}\right) \qquad\qquad 8\,\text{trivalent vertices.} \qquad (53)$$

The deltoidal icositetrahedron can be projected onto a cube so that each kite becomes one of four squares on the face of the cube. As a result, building the cage consists in placing a square tent on each face of a cube so that the bottom of the faces are joined with the two faces merging at the corners of the cube. The tent can be twisted.

We place the corner of the cube at the coordinates $(0, \pm 1, \pm 1/\sqrt{2})$ and $(\pm 1, 0, \pm 1/\sqrt{2})$ and pick $f_1 = (0, -1, 1/\sqrt{2})$. We take $v_1$ and $v_2$ as the orthonormal coordinate vectors for the reference face and $V$ as the center of the face:

$$\begin{aligned}
v_1 &= (\cos(\psi), -\sin(\psi)\sin(\theta), \sin(\psi)\cos(\theta)), \\
v_2 &= (0, \cos(\theta), \sin(\theta)), \\
V &= S\cos(\psi)(\sin(\theta) + \cos(\theta))(v_1 \times v_2).
\end{aligned} \qquad (54)$$

Then:

$$\begin{aligned}
W^1 &= R_{f_1}\!\left(\frac{2\pi}{3}\right)V, & w_1^1 &= R_{f_1}\!\left(\frac{2\pi}{3}\right)v_1, & w_2^1 &= R_{f_1}\!\left(\frac{2\pi}{3}\right)v_2 \\
W^3 &= R_z\!\left(\frac{\pi}{2}\right)V, & w_1^3 &= R_z\!\left(\frac{\pi}{2}\right)v_1, & w_2^3 &= R_z\!\left(\frac{\pi}{2}\right)v_2
\end{aligned} \qquad (55)$$

together with the corresponding $U^1$, $u^1$, $U^3$, and $u^3$.

The line $n_1 - n_2$ corresponds to the intersection of the master plane and its $2\pi/3$ rotation around the $f_1$ axis. Then $n_3$ and $n_4$ are obtained respectively by rotating $n_2$ and $n_1$ by $-2\pi/3$ around the $f_1$ axis.

The line $n_5 - n_6$ corresponds to the intersection of the master plane and its $\pi/2$ rotation around the $z$ axis. Then $n_7$ and $n_8$ are obtained by respectively rotating $n_6$ and $n_5$ by $-\pi/2$ around the $z$ axis.

$$\begin{aligned}
n_1 &= U^1 + t_1 u^1, & n_2 &= U^1 + t_2 u^1, & n_3 &= R_{f_1}\!\left(\frac{2\pi}{3}\right)n_2, & n_4 &= R_{f_1}\!\left(\frac{2\pi}{3}\right)n_1, \\
n_5 &= U^3 + t_5 u^3, & n_6 &= U^3 + t_6 u^3, & n_7 &= R_z\!\left(\frac{-\pi}{2}\right)n_6, & n_8 &= R_z\!\left(\frac{-\pi}{2}\right)n_5.
\end{aligned} \qquad (56)$$

To ensure that the cage is convex, we must also impose that the triangular tents are not concave. This implies the following condition: $(n \times f_1) \cdot v_1 > 0$.

The optimizing variables are $S$, $\psi$, $t_1$, $t_2$, $t_5$, $t_6$, Equations (54) and (56), as well as the coordinates, in the face plane, of the non shared vertices (Figure 12).

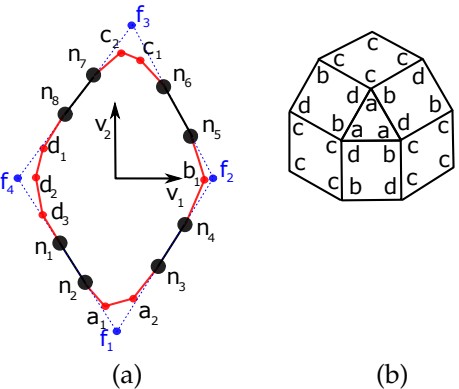

$$(a) \qquad\qquad (b)$$

**Figure 12.** (**a**) Parametrization of the rhombicuboctahedron p-cage face. (**b**) Distribution of hole-edges.

### 8.5. Rhombicosidodecahedron

The dual of the rhombicosidodecahedron is the deltoidal hexecontahedron. It is made out of sixty kite faces. Three of the vertices are pentavalent while the fourth one is trivalent. The p-cage built from the rhombicosidodecahedron can be obtained by placing tents on the faces of a dodecahedron (Figure 13).

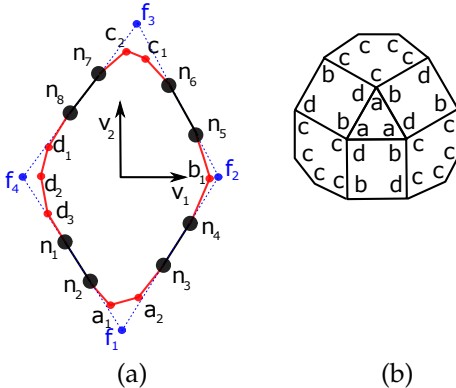

$$(a) \qquad\qquad (b)$$

**Figure 13.** (**a**) Parametrization of the rhombicosidodecahedron p-cage face. (**b**) Distribution of hole-edges.

If $a$ is the length of each edges, the inner radius of a dodecahedron is $r_i = a/2\sqrt{5/2 + 11\sqrt{5}/10}$, while the outer radius of each face is $r_h = a/(2\sin(\pi/5))$. Then the angle $\xi_d$ between the center of the face and one of the vertices is such that:

$$\tan(\xi_d) = \frac{r_h}{r_i} = \left( \sin(\frac{\pi}{5})\sqrt{\frac{5}{2} + \frac{11}{10}\sqrt{5}} \right)^{-1}. \tag{57}$$

To describe the plane of the reference face we take $V = (0, 0, 1)$ and define:

$$f_1 = (0, \tan(\xi_d), 1). \tag{58}$$

Then the orthonormal coordinate vectors for the master face $v_1$, $v_2$ are, once again, given by (54).

$$V = S\cos(\psi)(\cos(\theta) - \sin(\theta))\tan(\xi_d)(v_1 \times v_2). \tag{59}$$

Then:

$$W^1 = R_{f_1}(\frac{2\pi}{3})V, \qquad w_1^1 = R_{f_1}(\frac{2\pi}{3})v_1, \qquad w_2^1 = R_{f_1}(\frac{2\pi}{3})v_2$$

$$W^3 = R_z(\frac{2\pi}{5})V, \qquad w_1^3 = R_z(\frac{2\pi}{5})v_1, \qquad w_2^3 = R_z(\frac{2\pi}{5})v_2 \qquad (60)$$

together with the corresponding $U^1$, $u^1$, $U^3$, and $u^3$.

The line $n_1 - n_2$ corresponds to the intersection of the master plane and its $2\pi/3$ rotation around the $f_1$ axis. Then $n_3$ and $n_4$ are obtained by respectively rotating $n_2$ and $n_1$ by $-2\pi/3$ around the $f_1$ axis.

The line $n_5 - n_6$ corresponds to the intersection of the master plane and its $2\pi/5$ rotation around the $z$ axis. Then $n_7$ and $n_8$ are obtained by respectively rotating $n_6$ and $n_5$ by $-2\pi/5$ around the $z$ axis.

$$n_1 = U^1 + t_1 u^1, \qquad n_2 = U^1 + t_2 u^1, \qquad n_3 = R_{f_1}(\frac{2\pi}{3})n_2, \qquad n_4 = R_{f_1}(\frac{2\pi}{3})n_1,$$

$$n_5 = U^3 + t_5 u^3, \qquad n_6 = U^3 + t_6 u^3, \qquad n_7 = R_z(\frac{-2\pi}{5})n_6, \qquad n_8 = R_z(\frac{-2\pi}{5})n_5. \qquad (61)$$

To ensure that the cage is convex, we must also impose that the triangular tents are not concave. This implies the following conditions: $(n \times f_1) \cdot v_1 > 0$.

The optimizing variables are $S$, $\theta$, $\psi$, $t_1$, $t_2$, $t_5$, $t_6$, Equations (59) and (61), as well as the coordinates, in the face plane, of the non shared vertices.

*8.6. Snub Cube*

The dual of the snub cube is the pentagonal icositetrahedron. It is made out of twenty-four irregular pentagons. It can be seen as six square base tents made out of irregular pentagons built around a regular cube. The main axes of these tents are the axes going through the centers of the faces of the cube. Alternatively, it can be seen as right triangular base tents, the main axes corresponding to the vertices of the cube (Figure 14).

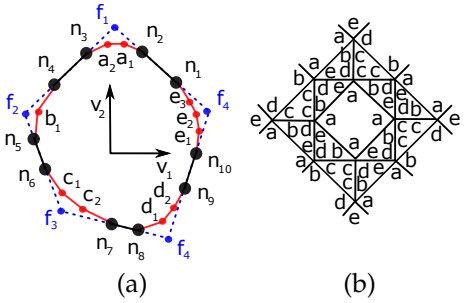

(a)                          (b)

**Figure 14.** (**a**) Parametrization of the snub cube p-cage face. (**b**) Distribution of hole-edges.

The vertices $n_3$ and $n_4$ are related respectively to the vertices $n_2$ and $n_1$ by a $-2\pi/4$ rotation around $f_1$, where $f_1$ points to the center of a face of a cube. The vertices $n_7$ and $n_8$ are related respectively to the vertices $n_6$ and $n_5$ by a $2\pi/3$ rotation around $f_3$ where $f_3$ points to a vertex of a cube. If we take $f_1 = (0,0,1)$ and $f_3 = (1,-1,1)$ then the vertices $n_9$ and $n_{10}$ are related to each other by a $\pi$ rotation around $g = (1,0,1)$.

We take:

$$v_1 = (1,0,0), \quad v_2 = (0,\cos(\theta),\sin(\theta)),$$

$$V = S\cos(\theta)(v_1 \times v_2) = S\cos(\theta)(0,-\sin(\theta),\cos(\theta)) \qquad (62)$$

We then rotate the vectors $f_3$ and $g_1$ by an angle $\psi$. For the pentagonal icositetrahedron $\theta \approx 31.75°$ $\psi \approx 61.46°$.

Using the vector $g = (1, 0, 1)$ we define:

$$W^1 = R_{f_1}(\frac{\pi}{2})V, \qquad w_1^1 = R_{f_1}(\frac{\pi}{2})v_1, \qquad w_2^1 = R_{f_1}(\frac{\pi}{2})v_2$$

$$W^2 = R_{f_3}(\frac{2\pi}{3})V, \qquad w_1^2 = R_{f_3}(\frac{2\pi}{3})v_1, \qquad w_2^2 = R_{f_3}(\frac{2\pi}{3})v_2$$

$$W^3 = R_g(\frac{2\pi}{3})V, \qquad w_1^3 = R_g(\pi)v_1, \qquad w_2^3 = R_g(\pi)v_2. \qquad (63)$$

together with the corresponding $U^i$ and $u^i$.

$$n_1 = U^1 + t_1 u^1, \qquad n_2 = U^1 + t_2 u^1, \qquad n_3 = R_{f_1}(\frac{-\pi}{2})n_2, \qquad n_4 = R_{f_1}(\frac{-\pi}{2})n_1,$$

$$n_5 = U^2 + t_5 u^2, \qquad n_6 = U^2 + t_6 u^2, \qquad n_7 = R_{f_3}(\frac{-2\pi}{3})n_6, \qquad n_8 = R_{f_3}(\frac{-2\pi}{3})n_5,$$

$$n_9 = U^3 + t_9 u^3, \qquad n_{10} = U^3 - t_9 u^3. \qquad (64)$$

The optimizing variables are $S$, $\theta$, $\psi$, $t_1$, $t_2$, $t_5$, $t_6$, $t_9$, Equations (62) and (64), as well as the coordinates, in the face plane, of the non shared vertices.

### 8.7. Snub Dodecahedron

The dual of the snub dodecahedron is the pentagonal hexecontahedron. It is made out of sixty irregular pentagons. It can be seen as twelve pentagonal base tents made out of irregular pentagons built around a regular dodecahedron. The main axes of these tents are the axes going through the centers of the faces or the dodecahedron. Alternatively it can be seen as twenty triangular base tents, the main axes corresponding to the vertices of the dodecahedron (Figure 15).

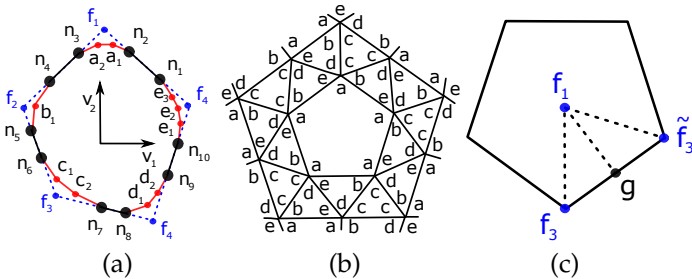

(a)       (b)       (c)

**Figure 15.** (**a**) Parametrization of the snub dodecahedron p-cage face. Base of the dodecahedron. (**b**) Distribution of hole-edges. (**c**) Vectors on the face.

The vertices $n_3$ and $n_4$ are related respectively to the vertices $n_2$ and $n_1$ by a $-2\pi/5$ rotation around $f_1$, where $f_1$ points to the center of a face of a dodecahdron. The vertices $n_7$ and $n_8$ are related respectively to the vertices $n_6$ and $n_5$ by $2\pi/3$ rotation around $f_3$ where $f_3$ points to a vertex of a cube. If we take $f_1 = (0, 0, 1)$ and $f_3 = (0, \tan(\xi_d), 1)$ then $\hat{f}_3 = (\sin(2\pi/5)\tan(\xi_d), -\cos(2\pi/5)\tan(\xi_d), 1)$ where $\xi_d$ is given by (57). The vertices $n_9$ and $n_{10}$ are related to each other by a $\pi$ rotation around $g$ where $g = (f_3 + \hat{f}_3)/2 = (\sin(2\pi/5)\tan(\xi_d)/2, -(1 + \cos(2\pi/5))\tan(\xi_d)/2, 1)$.

We take:

$$v_1 = (1, 0, 0), \quad v_2 = (0, \cos(\theta), \sin(\theta)),$$

$$V = S\cos(\theta)(v_1 \times v_2) = S\cos(\theta)(0, -\sin(\theta), \cos(\theta)) \qquad (65)$$

Next, we rotate the vectors $f_3$ and $g_1$ by an angle $\psi$. For the pentagonal icositetrahedron $\theta \approx 31.75°$ $\psi \approx 61.46°$.

Using the vector $\boldsymbol{g} = (1, 0, 1)$ we define:

$$W^1 = R_{f_1}(\frac{2\pi}{5})V, \qquad w_1^1 = R_{f_1}(\frac{2\pi}{5})v_1, \qquad w_2^1 = R_{f_1}(\frac{2\pi}{5})v_2$$

$$W^2 = R_{f_3}(\frac{2\pi}{3})V, \qquad w_1^2 = R_{f_3}(\frac{2\pi}{3})v_1, \qquad w_2^2 = R_{f_3}(\frac{2\pi}{3})v_2$$

$$W^3 = R_g(\frac{2\pi}{3})V, \qquad w_1^3 = R_g(\pi)v_1, \qquad w_2^3 = R_g(\pi)v_2. \qquad (66)$$

together with the corresponding $\boldsymbol{U}^i$ and $\boldsymbol{u}^i$.

$$n_1 = \boldsymbol{U}^1 + t_1\boldsymbol{u}^1, \qquad n_2 = \boldsymbol{U}^1 + t_2\boldsymbol{u}^1, \qquad n_3 = R_{f_1}(\frac{-2\pi}{5})n_2, \qquad n_4 = R_{f_1}(\frac{-2\pi}{5})n_1,$$

$$n_5 = \boldsymbol{U}^2 + t_5\boldsymbol{u}^2, \qquad n_6 = \boldsymbol{U}^2 + t_6\boldsymbol{u}^2, \qquad n_7 = R_{f_3}(\frac{-2\pi}{3})n_6, \qquad n_8 = R_{f_3}(\frac{-2\pi}{3})n_5,$$

$$n_9 = \boldsymbol{U}^3 + t_9\boldsymbol{u}^3, \qquad n_{10} = \boldsymbol{U}^3 - t_9\boldsymbol{u}^3. \qquad (67)$$

The optimizing variables are $S$, $\theta$, $\psi$, $t_1$, $t_2$, $t_5$, $t_6$, $t_9$, Equations (65) and (67), as well as the coordinates, in the face plane, of the non shared vertices.

## 9. Results

We have found 2371 symmetric p-cages (near-miss or regular) for polygonal faces with 6 to 20 edges and a deformation not exceeding 10%. The least iregular p-cages are shown in Tables 6 and 7. The numbers of p-cages for each regular solid are listed in Table 8. The full description of all the p-cages with a deformation not exceeding 10% is listed in the Supplementary Materials. The Supplementary Materials also contains the coordinates of each p-cage as off files. In [7], p-cages were constructed without assuming the regularity of the faces, which could then be deformed differently from each others (P was ranging from 6 to 17 only). This potentially allows for less deformed p-cages than the symmetric ones. Most of the symmetric p-cages found here are similar to the p-cages found in [7] but there are a number of exceptions. A full comparison of the p-cages found here and in [7] is given in the Supplementary Materials. Sometimes the symmetric p-cages are slightly less deformed than the non-symmetric ones. This is due to the fact that the relaxation performed in [7] did not always find the least deformed p-cages. There are also examples where the non-symmetric p-cages are less deformed than the symmetric ones and close inspection of the faces of the non-symmetric p-cages shows that the faces are not identical, allowing for less deformed p-cages.

Most of the p-cages generated from the prisms and antiprisms look like two rings of faces joined together at the base, as shown in Table 6. Because the Platonic solids are dual of each other, the p-cages generated from them correspond to polygons placed on the faces of the dual solids of the hole-edge polyhedra. For the tetrahedron, the solid can be elongated while preserving the symmetry of the p-cage. This is also possible for the cube and the octahedron, but the corresponding p-cages are the ones built from the square prism and the square antiprism, respectively .

The p-cages generated from the Archimedean solids are overall more interesting. Notice that the only p-cages where the faces have five neighbors, the maximum allowed geometrically, are the p-cages generated from the snub cube and the snub dodecahedron. `Asc_P11_2_1_1_1` (Table 7) is the least deformed one, with a deformation below 0.5%. All the other p-cages with five neighbors have deformations in excess of 2% (there are no regular p-cages with five neighbors).

Some of the Archimedean p-cages have a mesh-like structure, such as `Atd_P17_2_6_6` for example. Others are more tightly packed, even when faces have only three neighbors, such as `Ati_P16_4_2_7` where some of the hole-edges come very close together, but without touching each other.

By imposing a subgroup of the Platonic group as a symmetry to the p-cages we have substantially reduced the number of parameters required to describe the most general configurations, hence making the optimization process much less CPU intensive. The resulting p-cages are quite aesthetic and we hope they will be of interest to designers or architects, especially as they have congruent faces. Imposing the symmetry upfront will also be important for future work where one can construct p-cages made out of two or more different types of polygonal faces. One will first have to identify planar graphs made out of two or more types of nodes and identify the ones for which all the nodes belonging to the same family are equivalent. Even with only two types of polygons, the number of possible p-cages will be much larger than what we have described in this paper.

**Table 6.** Least irregular p-cages built from prisms and antiprisms solid hole polyhedra.

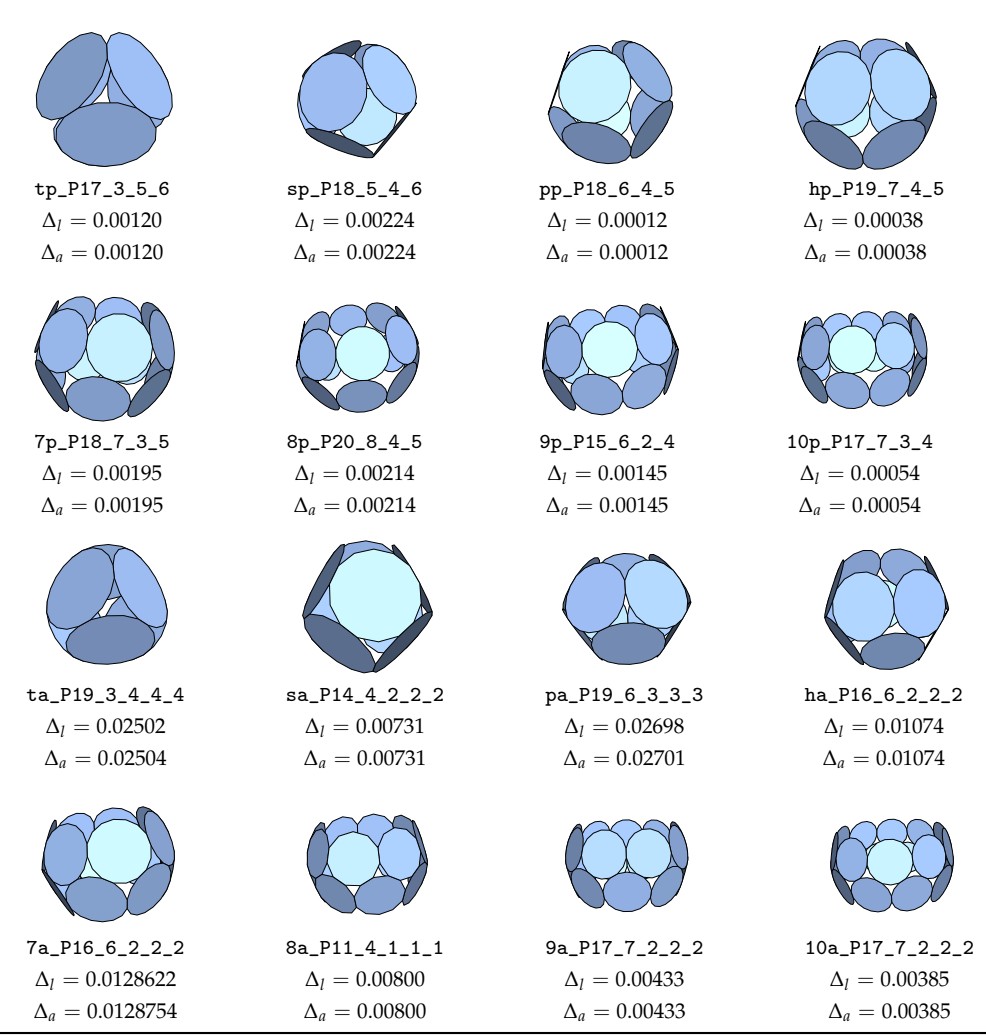

| tp_P17_3_5_6 | sp_P18_5_4_6 | pp_P18_6_4_5 | hp_P19_7_4_5 |
|:---:|:---:|:---:|:---:|
| $\Delta_l = 0.00120$ | $\Delta_l = 0.00224$ | $\Delta_l = 0.00012$ | $\Delta_l = 0.00038$ |
| $\Delta_a = 0.00120$ | $\Delta_a = 0.00224$ | $\Delta_a = 0.00012$ | $\Delta_a = 0.00038$ |
| 7p_P18_7_3_5 | 8p_P20_8_4_5 | 9p_P15_6_2_4 | 10p_P17_7_3_4 |
| $\Delta_l = 0.00195$ | $\Delta_l = 0.00214$ | $\Delta_l = 0.00145$ | $\Delta_l = 0.00054$ |
| $\Delta_a = 0.00195$ | $\Delta_a = 0.00214$ | $\Delta_a = 0.00145$ | $\Delta_a = 0.00054$ |
| ta_P19_3_4_4_4 | sa_P14_4_2_2_2 | pa_P19_6_3_3_3 | ha_P16_6_2_2_2 |
| $\Delta_l = 0.02502$ | $\Delta_l = 0.00731$ | $\Delta_l = 0.02698$ | $\Delta_l = 0.01074$ |
| $\Delta_a = 0.02504$ | $\Delta_a = 0.00731$ | $\Delta_a = 0.02701$ | $\Delta_a = 0.01074$ |
| 7a_P16_6_2_2_2 | 8a_P11_4_1_1_1 | 9a_P17_7_2_2_2 | 10a_P17_7_2_2_2 |
| $\Delta_l = 0.0128622$ | $\Delta_l = 0.00800$ | $\Delta_l = 0.00433$ | $\Delta_l = 0.00385$ |
| $\Delta_a = 0.0128754$ | $\Delta_a = 0.00800$ | $\Delta_a = 0.00433$ | $\Delta_a = 0.00385$ |

**Table 7.** Least irregular p-cages built from Platonic and Archimedean solid hole polyhedra.

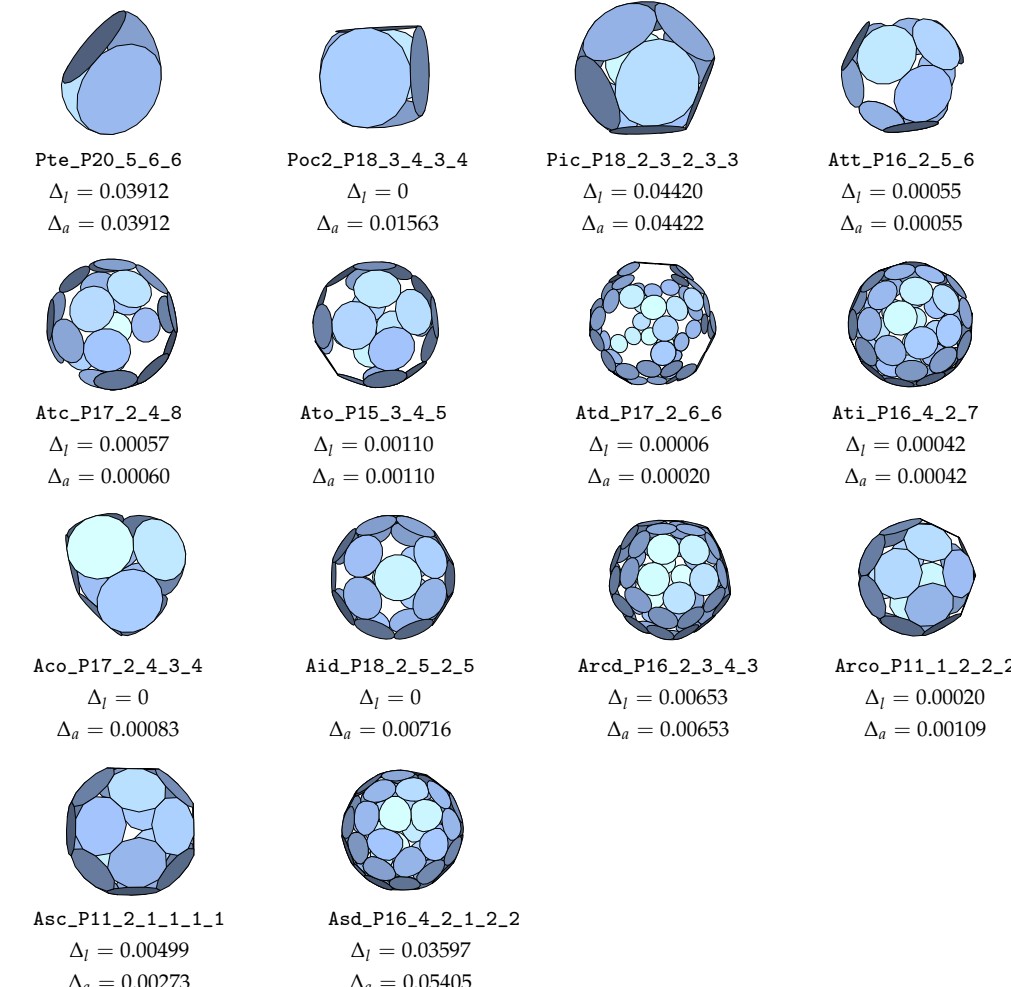

**Table 8.** Number of p-cages for each hole-edge regular solid. P = 6 to 20. $n$ is the total number of symmetric p-cages with a deformation smaller or equal to 10%, including regular p-cages. $n_r$ is the total number of regular p-cages (no deformations).

| Hole-Edge Solid | $n$ | $n_R$ | Hole-Edge Solid | $n$ | $n_R$ |
|---|---|---|---|---|---|
| Triangular Prism | 140 | 39 | Tetrahedron | 18 | 5 |
| Square Prism | 125 | 28 | Cube=Square Prism | (125) | (28) |
| Pentagonal Prism | 114 | 25 | Octahedron | 11 | 4 |
| Hexagonal Prism | 107 | 21 | Dodecahedron | 5 | 5 |
| Heptagonal Prism | 101 | 16 | Icosahedron | 16 | 3 |
| Octagonal Prism | 92 | 15 | Truncated Tetrahedron | 108 | 9 |
| Nonagonal Prism | 86 | 13 | Truncated Cube | 103 | 2 |
| Decagonal Prism | 80 | 13 | Truncated Octahedron | 122 | 7 |
| Triangular Antiprism | 42 | 4 | Truncated Dodecahedron | 102 | 1 |
| Square Antiprism | 31 | 0 | Truncated Icosahedron | 124 | 1 |
| Pentagonal Antiprism | 29 | 3 | Cuboctahedron | 193 | 4 |
| Hexagonal Antiprism | 26 | 0 | Rhombicuboctahedron | 232 | 5 |
| Heptagonal Antiprism | 24 | 0 | Snub Cube | 69 | 0 |
| Octagonal Antiprism | 23 | 0 | Snub Dodecahedron | 10 | 0 |
| Nonagonal Antiprism | 21 | 0 | Icosidodecahedron | 19 | 0 |
| Decagonal Antiprism | 20 | 0 | Rhombicosidodecahedron | 178 | 1 |

**Supplementary Materials:** The following supporting information is available in https://www.mdpi.com/article/10.3390/sym15030717/s1: It contains the files sym_pcages_sup_mat.pdf : Derivation of the possible symmetries of p-cages with Platonic and Archimedean hole-polyhedra; a list of all symmetric p-cage (P = 6 to 20) and a comparison between general p-cages and the symmetric ones. off_files.tar.gz : coordinates of all the p-cages as off files.

**Author Contributions:** B.M.A.G.P. derived the methods to analyze the near-miss symmetric p-cages and wrote the software to minimize the irregularity of the p-cage faces. Á.L. performed the group theory analysis of the hole polyhedron graphs and wrote the software to determine the constraints on the distribution of hole-edges. Conceptualization, B.M.A.G.P.; formal analysis, B.M.A.G.P.; funding acquisition B.M.A.G.P.; investigation, B.M.A.G.P., Á.L.; methodology, B.M.A.G.P., Á.L.; resources, B.M.A.G.P.; software, B.M.A.G.P., Á.L.; writing—original draft, B.M.A.G.P., Á.L. All authors have read and agreed to the published version of the manuscript.

**Funding:** This research was funded by the Leverhulme Trust Research Project Grants RPG-2020-306.

**Data Availability Statement:** The software used to determine the coordinates of the least deformed p-cages is available from zenodo: https://zenodo.org/record/7602763#.Y9z6O63P1aQ, doi:10.5281/zenodo.7602763 (Last accessed 8 March 2023). The software to determine the distribution of hole-edges on the planar graphs is available from zenodo: https://zenodo.org/record/7616018, doi:10.5281/zenodo.7616018. (Last accessed 8 March 2023) The coordinates of all the near-miss p-cages with deformations below 10% are available as off files from zenodo: https://zenodo.org/record/7602851#.Y90Ci63P1aQ, doi:10.5281/zenodo.7602851. (Last accessed 8 March 2023).

**Acknowledgments:** The p-cage figures where generated using geomview from www.geomview.org accessed on 8 March 2023.

**Conflicts of Interest:** The authors declare no conflict of interest.

## Abbreviations

The following abbreviation is used in this manuscript:

| | |
|---|---|
| TRAP | trp RNA-binding attenuation protein. |
| RNA | Ribonucleic acid: a nucleic acid present in all living cells. |
| DNA | Deoxyribonucleic acid: a nucleic acid present in all living cells. |

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
