# Peer review of "Near-Miss Symmetric Polyhedral Cages"

_symmetry, doi:10.3390/sym15030717_

Round 1
Reviewer 1 Report
See the attached file.

Reviewer 2 Report
Dear Authors,
throughout the manuscript, there appear some mistakes with notations, so it must be checked again. Phrases are clumsy and derogate the manuscript in general. You must consult the good English teacher friend if you want your manuscript to be noble.
Actions that you noted as simple examples are not simple at all without Figures, so if you want the manuscript to be clearly available to wide readers, then you need to put more Figures.
The crucial term called a hole-edge is not recognizable in Figure 1. Even, maybe, in Tables 6 and 7. Some Figures drawings (Figure 2) are a little clumsy.
Formula (2) is incorrect. Throughout the manuscript there are values undefined. Since that I can’t recommend your work to be published without revising it again.
All my comments I attached below. I am waiting for improvement to see if things will be better. If it possible, try to write the scalar product with a central dot:
(u⋅v)

Reviewer 3 Report
This paper defines the concept of homogeneous symmetric congruent equivalent near-miss polyhedral cages 2 made out of P-gons.
1) the main contribution and originality should be explained in more detail, which part of the proposal is new?
2) discussion of related works should be expanded with more recent works to help better situate the contribution. Add more paper from 2021 and 2022.
3) Minor grammar and syntax issues need correction to enhance readability
4)the conclusions should be extended with more discussion of future works.
Round 2
Reviewer 1 Report
Authors addressed the partial comments and they think literature review is enough. But according to me as an expert, litarture review should be improved. Therefore, I asked them to address my comments. Moreover, add recent work.
Therefore, my final decision is minor revision.
Reviewer 2 Report
Dear Authors,
the hole-edge is still not clear completely, but it is up to a reader that manage it. If you know-how, please, make it more clear.
Try to color in orange what the hole-edges are in Figure 1.

Reviewer 3 Report
The revised version can be accepted for publication.
